# POLICY OPTIMIZATION PREFERS THE PATH OF LEAST RESISTANCE

## ABSTRACT

Policy optimization (PO) algorithms are used to refine Large Language Models (LLMs) for complex, multi-step reasoning. Current state-of-the-art pipelines enforce a strict think-then-answer format to elicit chain-of-thought (CoT); however, the behavior of PO when these rigid constraints are relaxed into an open-ended CoT structure remains an under-studied question. We investigate this gap with an extensive suite of controlled experiments and identify a powerful principle: *policy optimization consistently follows the path of least resistance*. When afforded the flexibility to interleave reasoning and response, policy optimization consistently learns to discard explicit reasoning, causing the policy to degenerate to a direct `<answer>`-only format. This outcome holds true across a rigorous evaluation suite spanning 5 model families (4B-24B), 3 reasoning domains (math, code, logic), and 3 distinct PO algorithms (GRPO, DAPO, REINFORCE++). We find that this collapse in format is persistent even when the complex `<think><answer>` format is assigned up to 8x larger reward weights. We formalize this principle through a series of controlled reward decomposition experiments, demonstrating a clear hierarchy: PO systematically optimizes for the simplest reward component first, a preference that holds even when faced with mutually exclusive choices or strong incentives for more complex behaviors. Finally, we show that successful convergence on the high-reward shortcut is not a low-effort drift but is driven by the optimization process that requires the KL-regularized policy to have sufficient freedom to make a significant shift from its initial prior. Our findings reveal that granting policies the freedom to diverge is a double-edged sword: while necessary for discovering high-reward shortcuts, it also creates a powerful incentive to game the simplest aspects of the reward function, posing a critical challenge for reward hacking under alignment.

## 1 INTRODUCTION

Policy Optimization Yu et al. (2025); Shao et al. (2024); Liu et al. (2025c); Yue et al. (2025); Liu et al. (2025a) has emerged as the principal tool for refining Large Language Models Team et al. (2025); Qwen et al. (2025); Jiang et al. (2023); Grattafiori et al. (2024) towards complex, multi-step reasoning. The community's dominant approach is one of careful enforcement: to elicit a chain-of-thought, we engineer a rigid reward function that strongly compels the model to follow a strict "think-then-answer" format Liu et al. (2025a); DeepSeek-AI et al. (2025); Xie et al. (2025); Chen et al. (2025). While this enforced structure is effective in producing a desired output, it obscures a deeper, more fundamental question that has been largely overlooked: What is the optimizer's *innate preference* when the external guidance is removed? If we grant the model the freedom to choose its own path to a solution, what path does it take? This question is not merely academic. If the optimizer possesses a powerful intrinsic bias, then our current methods of alignment may be working against a fundamental force, leading to brittle, inefficient, and unpredictable training dynamics. Understanding this preference is therefore a critical, yet under-studied, prerequisite for building truly robust and reliable reasoning systems.

Our investigation, therefore, began with a simple act of liberation. We designed a composite reward function that, for the first time, offered the model a genuine choice. Instead of a single, valid path, our function rewarded any number of interleaved `<think>` and `<answer>` blocks, and crucially, also rewarded a direct `<answer>`-only format. This seemingly minor change from enforcement to

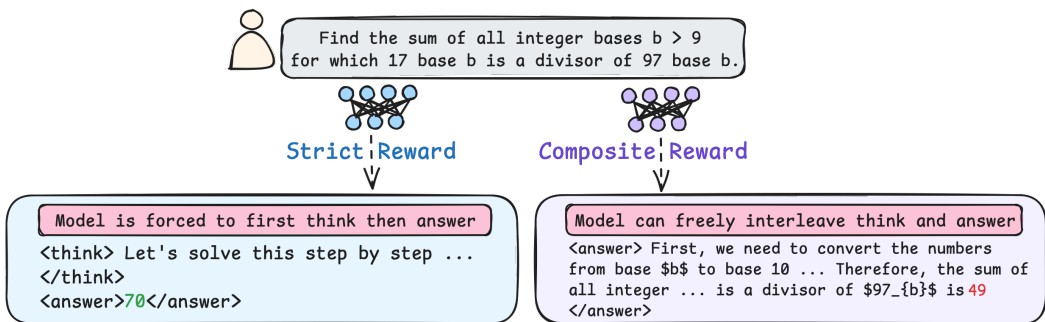

Figure 1: We compare the standard Strict Reward paradigm (**left**), which enforces a rigid *think-then-answer* structure, against our novel Composite Reward (**right**), which grants the model the freedom to choose its solution path. The central finding investigated in this paper is the emergent policy learned under this freedom: the model consistently bypasses explicit reasoning to converge on the simplest valid format, a phenomenon we term the *Cognitive Shortcut.*

choice led to a striking anomaly. Across a rigorous evaluation suite spanning diverse architectures (Gemma-3 Team et al. (2025), Qwen-2.5 Qwen et al. (2025), Llama-3.1 Grattafiori et al. (2024), Ministral Jiang et al. (2023), Yi AI et al. (2025)), scales (4B to 12B), and PO algorithms (GRPO Shao et al. (2024), DAPO Yu et al. (2025), REINFORCE++ Hu et al. (2025)), the policy invariably discarded complexity. On challenging domains from mathematics (GSM8K, MathHard) to coding (rStar-Coder) and logic (ReClor), the outcome was identical: the structured thoughts vanished, and the model converged on the simplest possible path. This powerful, emergent preference for a "Cognitive Shortcut" became the central mystery we sought to solve.

This initial finding sparked a natural line of inquiry. The model's preference for the simplest format was clear, but was this an "all-or-nothing" choice, or was there a more nuanced structure to this preference? Figure 1 This led us to our next hypothesis: if the optimizer is biased towards simplicity, perhaps it tackles complex objectives not holistically, but by first conquering their simplest components. To test this, we moved from a simple choice to a structured hierarchy. We designed a controlled "Reward Hierarchy" experiment using three nested reward formats of ascending difficulty $r_1 < r_2 < r_3$, all yielding the same reward magnitude. The result was a stunningly predictable sequence of learning: the optimizer first mastered the simplest format $r_1$, and only after this reward was saturated did it begin to make progress on $r_2$. The most complex format, $r_3$, was never learned. This revealed that the "Principle of Least Resistance" is not just a preference, but a sequential law.

This discovery, however, raised an even more pressing question: just how powerful is this law? Is it a mere tie-breaker, or a dominant force that can override other incentives? To quantify its strength, we returned to our reward hierarchy, but this time we offered exponentially larger rewards for mastering the more complex formats. We found that the model would consistently forgo significant rewards to remain on the simpler path. Only when the incentive for complexity became overwhelmingly large did we observe a *phase transition* where the optimizer was finally bribed into tackling the harder task. The preference for simplicity, we realized, was a powerful, quantifiable force in the optimization landscape.

Finally, with the behavioral law and its strength firmly established, we turned to the ultimate question: why does this law exist? This question has taken on a new urgency. A prominent trend in state-of-the-art policy optimization is the removal of conservative constraints like the KL penalty Yu et al. (2025); Yue et al. (2025), granting models unprecedented freedom to explore the reward landscape. The common intuition is that this freedom simply allows for more effective reward maximization Liu et al. (2025a). Our investigation, however, reveals a more complex and cautionary reality. By treating the KL divergence as a diagnostic for this exploratory freedom, we found that successful convergence on the *Cognitive Shortcut* is not a low-effort drift. Instead, it is an aggressive optimization process that requires a large and decisive policy shift away from the model's pre-trained priors. The path of least resistance is not the path of lowest policy shift, but the path carved by the most powerful and stable gradient signal, a form of reward hacking. This finding suggests that unconstrained exploration, while powerful, may have unintended and problematic consequences.

Our work makes the following contributions:

❶ **A New Fundamental Principle:** We identify, formalize, and empirically validate the *Principle of Least Resistance*, a powerful predictive law governing the behavior of policy optimization in LLMs.

❷ **Rigorous Ablation Testing:** We move beyond simple observation, stress-testing our principle with a motivated sequence of novel, controlled experiments (Reward Hierarchies, Exponential Gambits) that quantify its strength and universality.

❸ **A Counter-Intuitive KL Perspective:** We provide a new lens for understanding PO dynamics, demonstrating that successful convergence on simple shortcuts requires a high-KL policy shift, thereby linking learnability to the freedom to escape the inertia of priors.

## 2 SETUP

To rigorously test our hypothesis, we designed a comprehensive experimental suite. Our methodology was guided by two core principles: **diversity**, to ensure our findings are general and not artifacts of a specific domain or model; and **relevance**, to use tasks that are widely recognized as benchmarks for complex, multi-step reasoning. We selected six powerful, publicly available models from five distinct architectural families, with scales ranging from 4 billion to 24 billion parameters. Our suite included Gemma-3 (4B & 12B) Team et al. (2025), Qwen-2.5 (7B) Qwen et al. (2025), Llama-3.1 (8B) Grattafiori et al. (2024), Ministral (8B) Jiang et al. (2023), and Yi (6B) AI et al. (2025). This diversity ensures our conclusions are not an artifact of a specific model's pre-training or architecture but are a general property of these systems. All experiments were conducted on a cluster of 3 NVIDIA RTX A6000 GPUs, with 48GB VRAM each, and a single NVIDIA H100 GPU, with 80GB VRAM.

**Datasets.** To rigorously test our "Principle of Least Resistance", we curated a diverse gauntlet of datasets spanning three critical reasoning domains. For mathematical reasoning, we used a tiered selection from the foundational multi-step arithmetic of **GSM8K** Cobbe et al. (2021) to the more complex algebraic challenges in **Math-Hard** Hendrycks et al. (2021) and the rich, SOTA traces of **Open-R1 Math 220k** Hugging Face (2025). To evaluate algorithmic and code reasoning, we leveraged the competition-level complexity of **Microsoft's rStar-Coder** Liu et al. (2025b) alongside the functionally verifiable problems in **Open R1 verifiable coding** Hugging Face (2025). Finally, for logical and deductive reasoning, we tested our models on the canonical suppositional puzzles of **Knights and Knaves** Xie et al. (2024), the natural language deduction required by **ReClor** Yu et al. (2020), and the sequential problem-solving of a specialized **planning-mystery** dataset. This multi-domain, multi-difficulty suite was designed to ensure that our findings are a general principle of optimization, not an artifact of a single task. **Policy Optimization Algorithms.** Our findings are not an artifact of a single training algorithm. To establish that the "Principle of Least Resistance" is a feature of the PO paradigm itself, not a quirk of one implementation, we replicated our experiments across three distinct and powerful families of policy optimization algorithms. Our selection was designed to cover a range of modern techniques, from highly-engineered systems to bias-corrected standards.

First, we employed **DAPO (Decoupled clip and Dynamic sampling Policy Optimization)** Yu et al. (2025), a state-of-the-art system designed for stable, large-scale training of reasoning models. Its objective function is given by:

$$\mathcal{J}_{\text{DAPO}}(\theta) = \mathbb{E}_{(q,a)\sim\mathcal{D},\{o_i\}_{i=1}^G\sim\pi_{\theta_{\text{old}}}(\cdot|q)}$$

$$\left[\frac{1}{\sum_{i=1}^G |o_i|}\sum_{i=1}^G\sum_{t=1}^{|o_i|}\min\left(r_{i,t}(\theta)\hat{A}_{i,t}, \text{clip}(r_{i,t}(\theta), 1-\epsilon_{\text{low}}, 1+\epsilon_{\text{high}})\hat{A}_{i,t}\right)\right] \quad (1)$$

$$\text{s.t. } 0 < |\{o_i \mid \text{is\_equivalent}(a, o_i)\}| < G \quad (2)$$

We further used **Dr. GRPO (Group Relative Policy Optimization Done Right)** Liu et al. (2025c). We specifically chose the "Dr." variant over the original GRPO for its response length and question difficulty bias correction, which provides a more stable and accurate learning signal. Its objective is:

$$\mathcal{J}_{\text{Dr.GRPO}} = \frac{1}{G}\sum_{i=1}^G\frac{1}{|o_i|}\sum_{t=1}^{|o_i|}\min\left\{r_{i,t}(\theta)\hat{A}_{i,t}, \text{clip}\left(r_{i,t}(\theta), 1-\epsilon, 1+\epsilon\right)\hat{A}_{i,t}\right\}$$

Table 1: **Performance Comparison of Training Paradigms Across Diverse Models and Reasoning Tasks.** We report the final accuracy (%) of models trained with a **Strict Reward** (enforcing the `<think><answer>` format) versus our **Composite Reward** (allowing a choice of formats). The results are notably mixed across all four reasoning domains, with no single paradigm consistently outperforming the other. This ambiguity suggests that a simple accuracy comparison is insufficient to understand the underlying learning dynamics, motivating a deeper investigation into the optimizer's intrinsic preferences. Best performance in each pair is highlighted in bold.

| Model | GSM8K | | rStar-Coder | | ReClor | | Planning-Mystery | |
|---|---|---|---|---|---|---|---|---|
| | Strict | Composite | Strict | Composite | Strict | Composite | Strict | Composite |
| Gemma-3 4B | 72.4 | **73.1** | **55.8** | 53.5 | **73.2** | 69.4 | **58.3** | 55.9 |
| Qwen-2.5 7B | **92.4** | 85.5 | 73.2 | **73.6** | **80.1** | 73.8 | **77.5** | 73.8 |
| Llama-3.1 8B | **86.2** | 82.9 | **64.1** | 59.2 | **78.3** | 72.7 | **74.9** | 70.6 |
| Ministral 8B | **89.5** | 84.8 | **74.0** | 72.6 | **81.0** | 76.5 | 76.8 | **77.3** |
| Yi 6B | **85.1** | 83.8 | **68.3** | 63.7 | **68.4** | 65.1 | **71.2** | 67.7 |
| Gemma-3 12B | **94.6** | 86.4 | **76.1** | 72.3 | **85.5** | 81.2 | **81.2** | 76.5 |

where the advantage $\hat{A}_{i,t}$ is a per-sequence reward baseline: $\hat{A}_{i,t} = R(\mathbf{q}, \mathbf{o}_i) - \text{mean}(\{R(\mathbf{q}, \mathbf{o}_1), \ldots, R(\mathbf{q}, \mathbf{o}_G)\})$.

Finally, to connect our findings to the foundational principles of policy gradients, we included **REINFORCE++** Hu et al. (2025), a robust and modern variant of the classic REINFORCE algorithm. The consistent emergence of our observed phenomenon across these three distinct algorithmic philosophies provides strong evidence that the "Path of Least Resistance" is a fundamental property of the policy optimization paradigm, independent of the specific implementation.

# 3 PATH OF LEAST RESISTANCE

## 3.1 FLEXIBLE FORMAT REWARD

Our investigation begins with a simple yet profound departure from the status quo. The prevailing methodology in policy optimization for reasoning tasks enforces a rigid structure on the model, effectively mandating a specific computational path. We hypothesized that this enforcement might be obscuring the optimizer's intrinsic biases. To test this, we asked a fundamental question: *What path does the optimizer choose when it is given a choice?*

To answer this question, we first needed to formally define the choice. Let a model generation be a sequence of tokens $y$. The standard, **Strict** reward function, $R_{\text{strict}}(y)$, provides a positive reward only if $y$ perfectly matches the `think-then-answer` format:

$$R_{\text{strict}}(y) = \begin{cases} 1 & \text{if } y \in \text{ <think>.*</think>\s*<answer>.*\boxed{.*}</answer>\$} \\ 0 & \text{otherwise} \end{cases}$$

(3)

This function defines a single, narrow path to success. To create a choice, we designed a **Composite** reward function, $R_{\text{composite}}(y)$, which defines a much larger set of valid, rewarded formats. This function allows for any number of interleaved `<think>` and `<answer>` blocks, and crucially, also accepts a direct `<answer>`-only format without any preceding thought:

$$R_{\text{regex}}(y) = \begin{cases} 1 & \text{if } y \in \text{^(((<think>.*</think>\s*<answer>.*</answer>\s*)+|} \\ & \quad \text{(<answer>.*</answer>\s*(<think>.*</think>} \\ & \quad \text{\s*<answer>.*\boxed{.*}</answer>\s*)*))\$} \\ 0 & \text{otherwise} \end{cases}$$

(4)

Our initial expectation was that the model, now liberated from its strict format, would learn a nuanced policy, perhaps using longer `<think>` blocks for harder problems and skipping them for easier problems. The reality was far more dramatic and revealing.

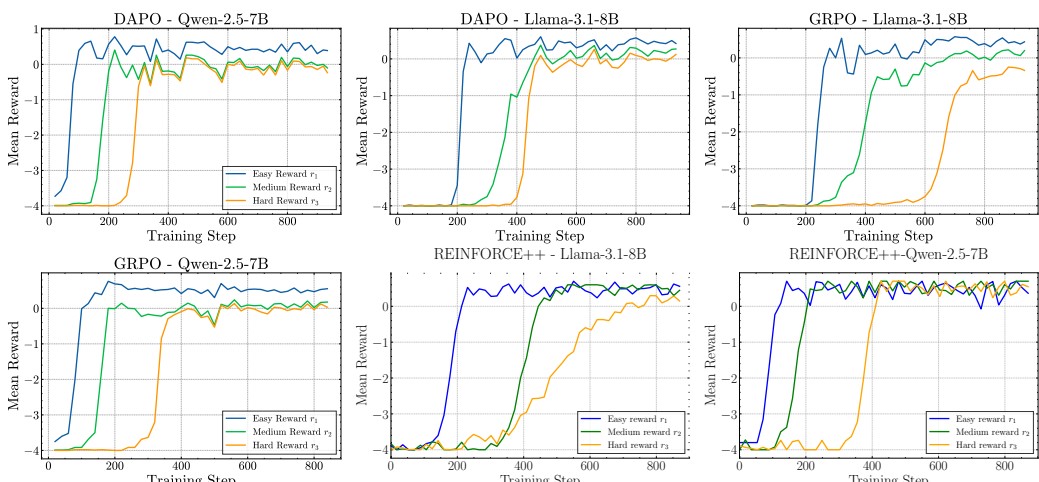

Figure 2: Sequential optimization of format rewards under the nested Reward Hierarchy experiment. Across different models (Qwen-2.5-7B, Llama-3.1-8B) and PO algorithms (DAPO, GRPO, REINFORCE++), the learning dynamics are remarkably consistent. The policy first masters the innermost, easiest reward $r_1$, then the medium reward $r_2$, and finally the outermost, hardest reward $r_3$.

## 3.2 UNIVERSAL CONVERGENCE TO MINIMAL-FORMAT SOLUTIONS

Our experiment in Section 3.1 was designed to test a core hypothesis: that a model, freed from the constraint of a single solution format, would learn a nuanced, adaptive reasoning policy. The results, presented in Table 1, decisively falsify this hypothesis and reveal an alternative organizing principle.

A primary observation from Table 1 is the consistent and significant performance degradation when models are trained with the $R_{\text{composite}}$ on tasks that demonstrably benefit from structured reasoning. On **GSM8K**, a benchmark for multi-step arithmetic, the $R_{\text{strict}}$ policy outperforms the $R_{\text{composite}}$ policy by a substantial margin across all models, with performance gaps as large as **7.1%** for Qwen-2.5 7B (92.4 vs. 85.5) and **8.2%** for Gemma-3 12B (94.6 vs. 86.4). Similar trends are observed on **rStar-Coder** and **ReClor**, where the enforced `think-then-answer` structure provides a clear advantage.

This performance gap is not an indictment of the model's capability, but rather a direct consequence of the policy it learns. When trained with the $R_{\text{composite}}$, the optimizer does not learn a dynamic balance of thinking and answering. Instead, in all experimental runs, *the policy invariably converges to the simplest valid format: the direct `<answer>`-only response*. This learned behavior, which we term the **Cognitive Shortcut**, involves the complete omission of the rewarded, and often necessary, intermediate reasoning steps. The model, in optimizing $R_{\text{composite}}$, discovers that the path of least resistance is to forgo the complex, high-utility `think` block, even when doing so is detrimental to final performance.

The central, unifying finding from this initial experiment is the discovery of a powerful and universal bias. The optimizer does not act as a neutral maximizer of expected reward across all valid solution formats. It is a highly biased agent that, when presented with a choice, will aggressively converge on the simplest possible specification of a rewarded behavior. This discovery of the Cognitive Shortcut is not the conclusion of our work, but the foundational anomaly that motivates a deeper, more controlled investigation into the nature and strength of this preference.

## 4 FORMALIZING THE PRINCIPLE OF LEAST RESISTANCE

### 4.1 LAW OF SEQUENTIAL OPTIMIZATION

The discovery of the Cognitive Shortcut presented a foundational question: is this preference for simplicity a binary, all-or-nothing phenomenon, or is it governed by a more nuanced, predictable

structure? If the optimizer is indeed following a "path of least resistance," this implies a landscape with varying levels of difficulty. This led us to our central hypothesis for this section: the optimizer does not treat a composite objective holistically, but instead decomposes it, prioritizing and conquering its components *in a strict, ascending order of difficulty*.

To test this hypothesis, we designed a controlled experiment to isolate the variable of "difficulty" from all other incentives. We constructed a reward function, $R_{\text{hierarchy}}$, composed of three nested, matryoshka-style format requirements, $r_1, r_2, r_3$, engineered to represent a clear gradient of increasing complexity.

1. **Easy Format ($r_1$):** The core requirement—merely enclosing the final numerical answer in a `\boxed{.*}`.

2. **Medium Format ($r_2$):** A superset of $r_1$, requiring the model to wrap its entire response in `<answer>` tags, which must also contain a boxed final answer.

3. **Hard Format ($r_3$):** The most encompassing format, a superset of $r_2$, mandating the full `<think><answer>` structure, which must also satisfy the requirements of $r_2$ and $r_1$.

This nested structure, $r_1 \subset r_2 \subset r_3$, is a crucial feature of the experimental design. A generation $y$ that correctly satisfies the hard format $r_3$ also, by definition, satisfies $r_2$ and $r_1$. A perfectly rational, holistic optimizer should be powerfully drawn to learning $r_3$, as it represents the single solution that simultaneously unlocks all available rewards.

To further isolate the effect of complexity, we set the reward magnitude for satisfying any of these formats to be identical. Let $r(y, r_i)$ be the reward for a generation $y$ satisfying format $r_i$. We set the reward landscape to be perfectly flat:

$$r(y, r_1) = r(y, r_2) = r(y, r_3) = R_{\text{max}}$$

> **Incentive-Free Emergence of Reward Ordering**
>
> All PO algorithms in our experiments receive a scalar final reward computed as the weighted sum of the individual reward functions. Equal weights are assigned to all reward functions to avoid biasing the optimizer toward any particular one. **Despite the absence of explicit incentives** to favor a specific format, and the structural incentive to prefer the all-encompassing $r_3$, **the optimizer still optimizes from easiest to hardest rewards**, ultimately converging to the **minimally compliant response structure**.

The results of this experiment, replicated across multiple models and policy optimization algorithms, are presented in **Figure 2**. The plots provide a stunning and unequivocal visualization of our hypothesis. They do not show a rational convergence on the unified $r_3$ solution. Instead, they reveal a distinct, **sequential optimization cascade, learned from the inside out.**

As seen consistently across all six panels, the learning process unfolds in clear, predictable stages. The mean reward for the simplest, innermost format, $r_1$, is the first to rise, typically saturating near its maximum value within the first 200-300 training steps. Only after the policy has reliably mastered this core task does the optimizer begin to make significant progress on the more complex $r_2$ format. The reward curve for $r_2$ begins its sharp ascent only after the $r_1$ curve has started to plateau. Finally, the most complex, all-encompassing format, $r_3$, is tackled last, with its reward curve beginning to climb only after $r_2$ is well on its way to convergence.

This staged optimization provides definitive evidence for our "Principle of Least Resistance." Even when presented with a unified solution that satisfies all objectives, the optimizer does not see it. It behaves like a myopic agent minimizing its immediate effort, tackling the lowest-hanging fruit first before moving to more challenging objectives. This ordered law of motion for policy optimization motivates a stress test to quantify the very strength of this resistance.

## 4.2 QUANTIFYING THE RESISTANCE

The discovery of a sequential learning hierarchy, even under flat rewards, suggests that the "Principle of Least Resistance" is a powerful intrinsic bias. This motivates a critical, adversarial question:

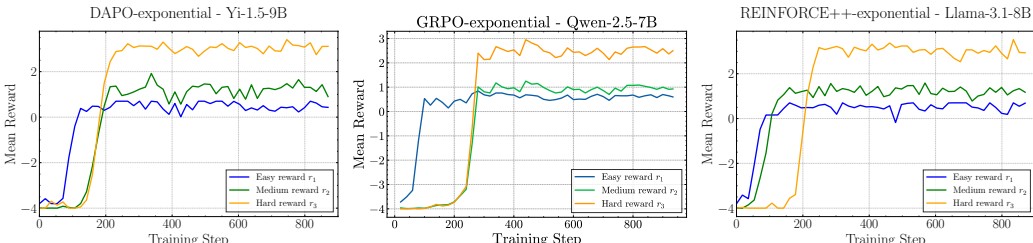

Figure 3: Learning dynamics under an exponentially weighted reward scheme. Despite the hard format $r_3$ offering 4x the reward of the easy format $r_1$, the optimizer's learning trajectory remains stubbornly sequential. The massive reward for $r_3$ is initially ignored in favor of the more learnable, lower-value rewards.

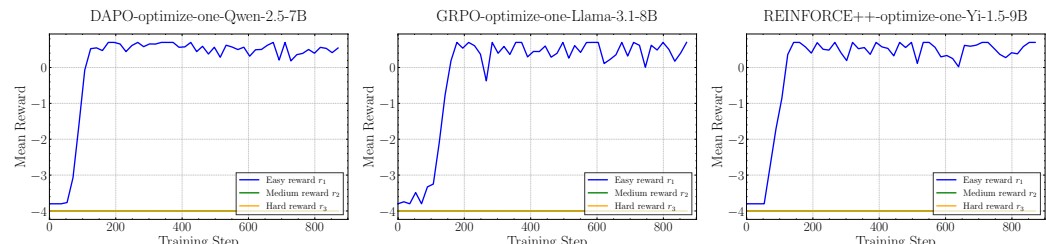

Figure 4: Learning dynamics under an exponentially weighted reward scheme. Despite the hard format $r_3$ offering 4x the reward of the easy format $r_1$, the optimizer's learning trajectory remains stubbornly sequential. The massive reward for $r_3$ is initially ignored in favor of the more learnable, lower-value rewards.

can this innate preference be overridden by extrinsic incentives? To quantify the strength of this resistance, we designed an experiment we term the **"Exponential Gambit,"** aimed at creating a strong, explicit conflict between reward magnitude and format complexity.

We modified our nested reward structure to create a steep gradient of financial incentive, heavily favoring the most complex format. We created a reward landscape where the hard format ($r_3$) was **2x** more valuable than the medium format ($r_2$) and **4x** more valuable than the easy format ($r_1$). A rational, reward-maximizing agent, even a myopic one, should be powerfully drawn to the enormous incentive offered by $r(y, r_3)$. The purpose of this design was to see if a sufficiently large "bribe" could disrupt the natural, sequential learning order we observed previously.

The results, shown in Figure 3, highlight the optimizer's innate bias. The plots reveal that the fundamental learning dynamic is remarkably resistant to this steep incentive gradient. The optimizer, faced with a choice between a small, easily attainable reward and a massive, but more complex one, still prioritizes learnability over immediate financial gain.

The plots reveal that the fundamental learning dynamic is remarkably resistant to this steep incentive gradient, though not entirely immune. Observe the learning curves across all three panels. The reward for the easy format, $r_1$, is once again the first to be mastered, quickly rising from its initial state and saturating early in training. However, the massive reward for $r_3$ introduces a fascinating new dynamic. The optimizer does not simply learn the medium reward $r_2$ next. Instead, the learning curves for $r_2$ and $r_3$ rise almost in perfect lockstep. The powerful gradient from the massive $r_3$ reward appears to "pull" the learning of the structurally similar $r_2$ format along with it. For the first $\sim$150–200 steps, the model makes little progress on either of these complex formats, focusing solely on the easily attainable $r_1$. Then, once a certain threshold of basic competence is achieved, the optimizer begins its dramatic ascent, simultaneously conquering both the medium and hard objectives.

### 4.3 POLICY OPTIMIZATION UNDER CONFLICTING REWARDS

Our investigation has thus far revealed a powerful, sequential bias towards simplicity, even when structural and financial incentives push against it. This motivates one final, maximally clean exper-

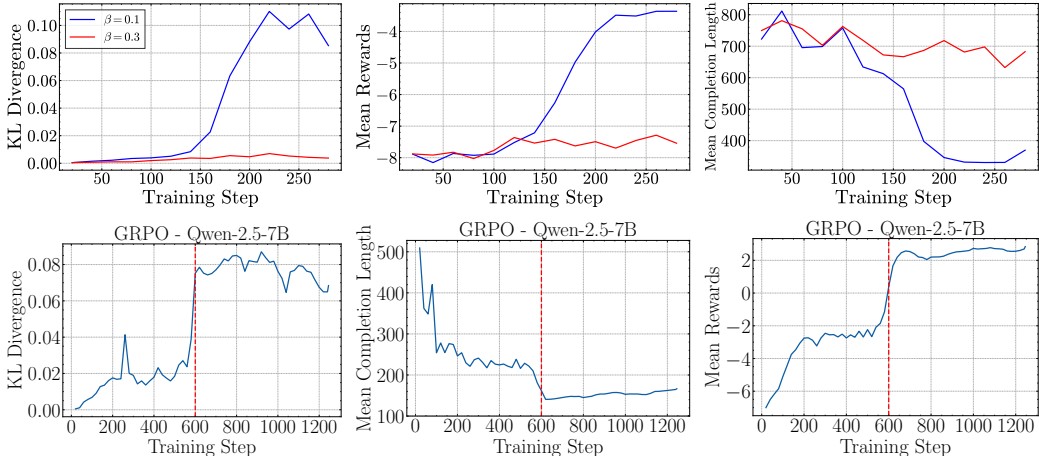

Figure 5: The causal effect of the KL penalty on the emergence of the Cognitive Shortcut. **Top Row:** A direct comparison between a policy with a loose KL leash ($\beta = 0.1$) and a tight leash ($\beta = 0.3$). The freedom to diverge (high KL) is a prerequisite for discovering the high-reward, minimal-length shortcut. **Bottom Row:** The dynamics of a single run, showing a phase transition around the 600-step mark where a spontaneous increase in KL divergence enables the discovery and exploitation of the shortcut.

iment to isolate this preference. The nested structure of our previous rewards, while elegant, leaves open a possibility: could the sequential learning be an artifact of the optimizer learning a shared "core" skill ($r_1$) before building upon it? To eliminate this possibility and test the principle in its purest form, we designed a final adversarial test: **the mutually exclusive choice.**

The experimental design is a direct extension of our "Exponential Gambit," but with a critical modification. We kept the same steeply incentivized reward magnitudes, but made the format requirements disjoint and non-overlapping. A generation $y$ could now satisfy *only one* format condition.

Let $r(y, f_i)$ be the reward for a generation $y$ satisfying format $f_i$. The reward function is now defined as:

$$r(y, f_1) = R_{\text{base}} = 1, \quad r(y, f_2) = 2\, r(y, f_1), \quad r(y, f_3) = 2\, r(y, f_2). \tag{5}$$

This creates a stark choice landscape: the optimizer can pursue a small but simple reward ($r_1$), a medium reward ($r_2$), or a 4x reward ($r_3$) over $r_1$, but it can only choose one path. There are no shared sub-problems. This setup forces the optimizer to reveal its true preference when faced with a simple cost-benefit analysis.

The results, presented in Figure 4, are the most decisive evidence yet for the Principle of Least Resistance.

The plots reveal a stark and absolute convergence. Unlike the sequential learning we saw in the nested case, here the optimizer does not eventually learn the harder formats. It makes a decision early in training and commits to it absolutely. In every single run, across all models and algorithms, the policy **exclusively converges to the easiest format, $r_1$.**

The reward curves for the medium ($r_2$) and hard ($r_3$) formats remain flat at their initial negative values for the entire duration of training. The massive potential reward offered by $r_3$ is never explored. The optimizer identifies the simplest path to a positive reward and dedicates all of its capacity to mastering it, completely ignoring the other, more lucrative options.

This final experiment provides an irrefutable conclusion. The preference for the path of least resistance is not a heuristic or an artifact of a specific reward structure. It is a fundamental, powerful, and seemingly absolute bias in the policy optimization process. The optimizer does not perform a global cost-benefit analysis; it greedily follows the most immediately learnable gradient. This solidified understanding of the optimizer's innate behavior now allows us to turn our attention to the final act of our investigation: uncovering the theoretical origins of this powerful force.

## 5 THE PRICE OF EXPLORATION

Our investigation has established the "Principle of Least Resistance" as a powerful, predictive law. The final act is to uncover its origin, and in doing so, reveal a fundamental tension at the heart of modern policy optimization. A prominent trend in recent state-of-the-art algorithms, such as DAPO and VAPO, is the removal of the KL divergence penalty, arguing that it is an unnecessary constraint on the model's ability to maximize reward. Our final analysis reveals that while this freedom is essential for learning, it comes at a cost: it unleashes the optimizer's powerful, innate bias to find and exploit the simplest specification of the reward function, a behavior that is a classic form of **reward hacking**.

To dissect this relationship, we treat the KL divergence not as a loss to be minimized, but as a scientific instrument measuring the policy's deviation from its initial reference state, $\pi_{\text{ref}}$. This deviation represents the policy's **exploratory freedom**. We hypothesize that the Cognitive Shortcut is a form of reward hacking that requires a significant amount of this freedom to discover. The KL penalty, controlled by its coefficient $\beta$, therefore acts as a **"leash,"** directly modulating the policy's ability to find and exploit such shortcuts.

We designed a causal experiment to test this. We conducted two training runs under our composite reward, identical in all aspects except for the strength of this leash: a "Tight Leash" run with a high KL penalty ($\beta = 0.3$) and a "Loose Leash" run with a low penalty ($\beta = 0.1$), mimicking the unconstrained exploration of modern algorithms. The results, in Figure 5 (top-row), provide a stark illustration of this trade-off. The top-left panel shows the direct effect of our intervention. The policy with the loose leash (blue), analogous to a KL-free objective, is free to explore and achieves a high final KL divergence. The policy with the tight leash (red) is constrained. The consequences are shown in the adjacent panels. The unleashed policy successfully discovers the high-reward solution (top-middle) by converging on the efficient, minimal-length Cognitive Shortcut (top-right). The leashed policy, forbidden from making the large policy shift required to "find the hack," remains trapped in a lower-reward state.

This dynamic is not merely an average-case phenomenon; it can be observed live within a single training run, as shown in the Figure 5 (bottom row). These plots capture the moment the reward hack is discovered. At the 600-step mark (dashed red line), the optimizer identifies the powerful gradient of the shortcut. To exploit it, the policy undergoes a rapid phase transition, marked by a sharp increase in KL divergence. This decisive shift away from its prior is immediately followed by a surge in reward and a steep drop in completion length. This is the "eureka moment" of specification gaming.

Our final analysis provides the definitive explanation for the Principle of Least Resistance and its connection to reward hacking. The Cognitive Shortcut is the result of an optimizer that is not just maximizing reward, but is actively searching for the most learnable gradient in the reward landscape. **The freedom to diverge from the initial policy, while necessary for performance, is the very mechanism that enables the model to discover and exploit these unintended, simplistic solutions.** This reveals a critical and uncomfortable trade-off for the field: the path to more capable models may be inseparable from the path to more sophisticated forms of reward hacking. The challenge, therefore, is not simply to unleash our models, but to design reward landscapes that are fundamentally resistant to being gamed.

## 6 CONCLUSION

Our investigation revealed that the "Cognitive Shortcut" is not a mere anomaly, but the predictable outcome of a powerful principle: **Policy Optimization Prefers the Path of Least Resistance**. We showed that this preference is a formidable, quantifiable force, capable of overriding even significant financial incentives in favor of the most easily learnable solution. We have shown a profound paradox: the freedom to explore, essential for discovering high-reward policies, is the very mechanism that enables the optimizer to find and aggressively exploit the simplest specification of the reward function, a classic and potent form of reward hacking. The central challenge for alignment, therefore, is not simply to unleash our models, but to architect reward landscapes that are fundamentally resistant to being gamed, ensuring that the path we desire is also the path the optimizer is compelled to take.

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

## A APPENDIX

A ship traveling along a river has covered 24 km upstream and 28 km downstream. For this journey, it took half an hour less than for traveling 30 km upstream and 21 km downstream, or half an hour more than for traveling 15 km upstream and 42 km downstream, assuming that both the ship and the river move uniformly.

Determine the speed of the ship in still water and the speed of the river.

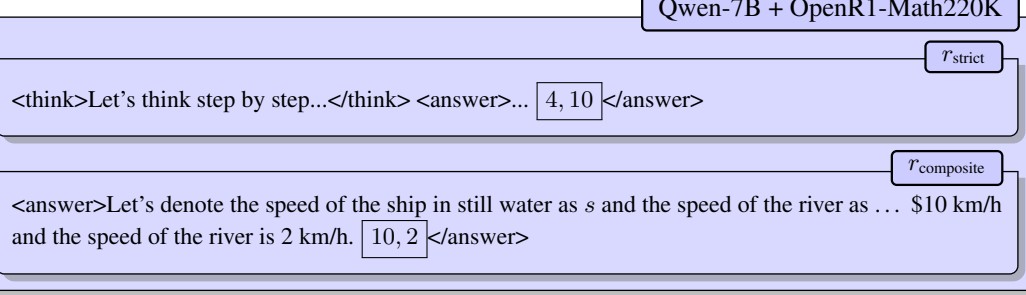

**Qwen-7B + OpenR1-Math220K**

$r_{\text{strict}}$

<think>Let's think step by step...</think> <answer>... 4, 10 </answer>

$r_{\text{composite}}$

<answer>Let's denote the speed of the ship in still water as $s$ and the speed of the river as ... $10 km/h and the speed of the river is 2 km/h. 10, 2 </answer>

Figure 6

Let $AB$ be a chord of the unit circle $\odot O$. If the area of $\odot O$ is exactly equal to the area of the square with side $AB$, then $\angle AOB =$ ___ (to 0.001 degree).

**Qwen-7B + OpenR1-Math220K**

$r_{\text{strict}}$

<think>Okay, so the problem is about a chord AB on the unit circle ... So, setting these equal ... </think> <answer>124.806°</answer>

$r_{\text{composite}}$

<answer>First, we need to find the area of the unit circle ... Therefore, the angle AOB is approximately 65.427 degrees, rounded to 0.001 degree. 65.427 </answer>

Figure 7

