# OpenReview forum: "Policy Optimization Prefers The Path Of Least Resistance"
_ICLR.cc/2026/Conference — ICLR 2026 Conference Withdrawn Submission_

### Official Review · Reviewer_6Qkg · 2025-10-30

**Soundness:** 1
**Presentation:** 2
**Contribution:** 2
**Rating:** 4
**Confidence:** 4

**Summary:**

This paper explores how Policy Optimisation (PO) algorithms behave when applied to Large Language Models (LLMs) that use open-ended Chain-of-Thought (CoT) reasoning. The authors propose a *Composite Reward* that lets the model choose between two response formats, a detailed reasoning style ($\langle \text{think} \rangle\langle \text{answer} \rangle$) or a concise, answer-only one. Their key finding, described as the *Principle of Least Resistance*, is that PO tends to converge toward the simplest possible solution, so-called *Cognitive Shortcut,* even when the more elaborate reasoning format is explicitly rewarded. They further connect this tendency to reward hacking, showing that models need sufficient freedom to explore (high KL divergence) before they can discover and exploit these shortcuts.

**Strengths:**

The *Principle of Least Resistance* is convincingly demonstrated across a broad range of setups, including five model families (4B–24B), three reasoning domains (math, code, logic), and three PO algorithms (GRPO, DAPO, REINFORCE++).

The analysis linking exploration freedom (high KL) to the discovery and exploitation of shortcuts offers an insightful perspective for diagnosing and mitigating reward hacking.

**Weaknesses:**

While the paper clearly establishes that the principle exists and what conditions enable it (high KL), the underlying mechanism, “why simpler rewards produce stronger and more stable gradients”, is only asserted, not deeply examined. The claim that these rewards provide the “most powerful and stable gradient signal” would be much stronger with a formal analysis or empirical comparison of gradient magnitudes and variances.

In Figure 2, the Reward Hierarchy experiments show that the complex reward $r_3$ is “never learned.” It would strengthen the conclusion to analyse the final policy composition more carefully: does the policy fully converge to $r_2$, or does it remain a mix of $r_1$ and $r_2$? A quantitative breakdown of the final mixture of reasoning formats would clarify this.

Table 1 shows that the Strict Reward often outperforms the Composite Reward on reasoning tasks. However, since $R_{\text{strict}}$ enforces a $\langle \text{think} \rangle$ block and $R_{\text{composite}}$ often collapses to an $\langle \text{answer} \rangle$-only mode, the comparison blends format preference with reasoning ability. The shortcut is problematic precisely because reasoning is required for hard problems. Showing the failure rate of $\langle \text{answer} \rangle$-only responses on such tasks would make the connection between shortcut behaviour and degraded performance more explicit.

**Questions:**

You claim the Cognitive Shortcut emerges because it provides the “most powerful and stable gradient signal.” Could you include empirical evidence for this—perhaps by comparing the average gradient magnitudes or variances between optimisation steps associated with $r_1$ and $r_3$?


For models that experience the greatest performance drop under $R_{\text{composite}}$, what fraction of *successful* outputs are pure $\langle \text{answer} \rangle$-only versus those that include $\langle \text{think} \rangle$ reasoning? Similarly, what fraction of *failed* outputs attempted reasoning but still failed? This quantitative breakdown would make the performance trade-off much clearer.

**Details Of Ethics Concerns:**

In addition, this paper doesn't include the LLM usage statement, and disobeys the LCLR rules

---

### Official Review · Reviewer_hEFT · 2025-11-01

**Soundness:** 2
**Presentation:** 1
**Contribution:** 2
**Rating:** 2
**Confidence:** 3

**Summary:**

This paper studies the behavior of policy optimization when the rigid ``think-then-answer'' format is relaxed in reasoning LLMs. The authors find that PO tends to converge to the simplest possible form (an <answer> only response) even when more complex reasoning structures yield higher rewards. They formalize this observation as the Principle of Least Resistance (PoLR) and validate it across multiple models, tasks, and algorithms (DAPO, GRPO, REINFORCE++). The paper concludes that unconstrained exploration (low KL penalty) facilitates this collapse and is tightly linked to reward hacking.

**Strengths:**

1. The paper demonstrates that PO systematically favors simpler, easier-to-learn behaviors when multiple valid reward paths exist.

2. Results are consistent across five LLM families, three reasoning domains, and three PO algorithms.

**Weaknesses:**

1. The method is conceptually simple but lacks technical innovation. The paper mostly reports empirical observations that may not generalize to all RL methods. For example, many works omit the format reward and only remain the accuracy reward as the reward function (e.g., Skywork OR1).

2. It would be interesting to see how the proposed method compares to LLM-as-a-judge.

3. There is no related work section in the main paper. Including it in the main text would provide essential context, clarify positioning relative to prior methods, and make the work more self-contained and complete.

**Questions:**

See above.

---

### Official Review · Reviewer_1uFE · 2025-11-02

**Soundness:** 1
**Presentation:** 2
**Contribution:** 1
**Rating:** 2
**Confidence:** 4

**Summary:**

The paper investigates what happens when policy optimization algorithms are given genuine freedom to choose solution formats for reasoning tasks. Rather than enforcing a rigid "think-then-answer" structure, they design a composite reward that accepts any number of interleaved reasoning/answer blocks or a direct answer-only format, and observe what models actually learn to produce. The authors conduct extensive controlled experiments across five model families (4B-24B parameters), three policy optimization algorithms (DAPO, Dr. GRPO, REINFORCE++), and three reasoning domains (math, code, logic). They progressively test more complex conditions: nested reward hierarchies with equal weights, exponentially weighted rewards favoring complexity, and mutually exclusive format choices. They measure what format emerges as well as KL divergence from the initial policy.

The authors claim that, across all tested conditions, models converge to the simplest valid format ("direct answers with no reasoning") despite this harming performance on reasoning tasks. Second, they identify a "Principle of Least Resistance" where policies optimize easier reward components sequentially before attempting harder ones. Third, even significant reward scaling toward complex formats only partially overrides this preference. Finally, they argue this convergence requires substantial KL divergence (policy deviation), suggesting it's active optimization toward a reward "hack" rather than passive drift, with potential implications for alignment under unconstrained exploration.

**Strengths:**

## Strength 1

The paper tests across five distinct model families (Gemma-3, Qwen-2.5, Llama-3.1, Ministral, Yi), spanning 4B to 24B parameters, and validates findings across three different policy optimization algorithms (DAPO, Dr. GRPO, REINFORCE++). This multi-axis validation significantly strengthens claims beyond a single-algorithm, single-model observation. Also, the paper uses established benchmarks (GSM8K, rStar-Coder, ReClor, planning mysteries) across three meaningful reasoning domains (math, code, logic).

## Strength 2

Beyond the core observation, the authors conduct a series of ablation studies to test nested rewards, exponentially-weighted rewards, and mutually exclusive rewards, helping to rule out some trivial explanations.

## Strength 3

Th epaper identifies a potentially interesting phenomenon with implications for how to structure rewards in practice or broader implications for safety post-training.

**Weaknesses:**

## Weakness 1

The paper only tests policy optimization algorithms for LLMs and it remains unclear whether this "preference for least resistance" generalizes to supervised fine-tuning, other RL algorithms, or other domains entirely. This severely limits the generality of the claimed "fundamental principle." In addition, the paper does not adequately address whether models collapse to simple answer formats because structured reasoning is simply harder to learn in sparse reward settings, rather than due to an intrinsic bias in PO. Binary rewards are notoriously difficult to optimize; this could just be standard local minima behavior rather than a special property of policy optimization. The binary reward function used throughout is extremely sparse. Mode collapse to simple solutions under sparse rewards is well-established in RL literature. The paper conflates this known phenomenon with a special discovery about PO, without adequately separating the two.

A critical question is whether these findings are artifacts of the authors' specific experimental design choices. Testing alternative reward architectures (continuous rewards, soft constraints, regularization toward CoT) would be essential to isolate whether the phenomenon is real or an artifact of the sparse, binary reward setup. Without this, it's unclear whether they discovered something about PO or just constructed a poorly-designed reward function.

## Weakness 2

The paper claims to identify a "fundamental principle" based purely on empirical observations across five models, yet provides no theoretical analysis or proofs. Calling this a "fundamental principle" requires either theoretical backing or much more extensive empirical validation than provided. The core observation is somewhat tautological once stated: of course optimization prefers easier paths.

While the controlled experiments are reasonably comprehensive, the paper entirely lacks theoretical analysis of why sequential difficulty-ordered optimization should occur. Providing even simple theoretical intuition (e.g., analyzing gradient geometry or optimization trajectories) would substantially strengthen the claims.

## Weakness 3

Why not test a composite reward that includes a small positive bonus for engaging with chain-of-thought reasoning? This simple ablation would reveal whether models are avoiding CoT due to its absence from rewards or due to active preference against it. The current design cannot distinguish between these.

## Weakness 4

The claim that intrinsic biases lead to "brittle, inefficient, and unpredictable" alignment is not adequately justified by the paper. Section 4.2 shows that sufficiently large reward incentives do partially override the preference, suggesting external guidance is not as fundamentally limited as claimed. The paper's claims about alignment and reward hacking in Section 5 are speculative and not sufficiently supported by the experiments. The connection from "models find simple rewards easier" to "this is a fundamental alignment threat" requires more rigorous argumentation and evidence beyond this specific scenario.

## Weakness 5

That policies learn easier objectives before harder ones is relatively unsurprising and well-known from curriculum learning research. The paper does not sufficiently explain why this observation specifically in their setup constitutes a surprising new principle worthy of ICLR publication.

In addition, using KL divergence to measure how far a policy deviates from its prior is not counter-intuitive, as it is literally the definition of what KL measures. Framing this as a "counter-intuitive KL perspective" appears to overstate the novelty of treating KL as a measure of exploratory freedom.

## Weaknes 6

Although the paper tests three PO algorithms (DAPO, Dr. GRPO, REINFORCE++), all are closely related policy gradient methods. Testing on more diverse optimization approaches (e.g., implicit methods, evolution strategies, or other paradigms) would better support claims of universality. Simialrly, while the paper tests six models, they are all relatively small (4B-24B parameters) and all contemporary. Testing on significantly larger models, older architectures, or models trained differently might reveal this is not universal.

## Weakness 7

The paper employs a dramatic and flowery writing style that feels more suited to popular science than rigorous research. Tone phrases like "powerful," "stunning," and "eureka moment" distract from scientific precision and suggest the authors may be overselling modest empirical findings.

## Weakness 8

The paper never explains the mechanism of why the simple answer-only format is easier to optimize than chain-of-thought. Is it easier to learn? Does it have a stronger gradient signal? Without understanding the "why," it's difficult to assess whether this is a special property of PO or just an artifact of their optimization landscape.

Could the result simply reflect that with a sparse binary reward, models overfit to the easiest way to obtain the positive signal? This is not distinguished from a more fundamental principle about PO and would suggest the finding is less significant than framed.

**Questions:**

I've embedded my questions in the listed weaknesses.

---

### Official Review · Reviewer_4i7v · 2025-11-03

**Soundness:** 1
**Presentation:** 2
**Contribution:** 2
**Rating:** 2
**Confidence:** 3

**Summary:**

The paper examines whether policy optimization (PO) tends to myopically search for easy but suboptimal reward channels. It compares RL accuracy when answers are formatted in a rigid "think-then-answer" style versus interleaved thinking/answering, reporting that Strict outperforms Composite in 21/24 model–dataset pairs. It also trains with indicator-style rewards for output formatting and observes that the corresponding tasks are learned in reverse-subset order (general to specific). This pattern persists when increasing the rewards on the more specific formats. With higher KL penalty, learning stalls. Finally, for three unspecified "disjoint" rewards, only the easy one is learned.

**Strengths:**

- The explicit inclusion of a regex in section 3.1 clarifies the target behavior (though the first regex appears to be missing a leading "^").
- The framing of rewarding formatting to probe basic optimization behavior is clean and targets core questions about PO dynamics.

**Weaknesses:**

**Scope/fit of Table 1.** Table 1 appears only loosely related to the claimed "easy-before-hard" phenomenon. It contrasts RL accuracies under simple vs. complex prompts, but the runs are presumably trained to convergence. Presenting R_strict and R_regex as "rewards" is confusing in this context: it is unclear whether these are multiplied by task correctness during training (and, from the wording, whether the model effectively chooses between them via a max over indicators).

**Underspecified "disjoint" rewards (Figure 4).** The three "disjoint" objectives are not defined (unless "disjoint" means recoding $(r_1, r_2, r_3)$ into $(r1, r2 \land \lnot r1, r3 \land \lnot r_1 \land \lnot r_2)$, which would be unusual). Without token- or regex-level specifications and without measuring shared substructure, the result is difficult to interpret.

# Core experiments do not isolate the core hypothesis.
The headline hypothesis is that easy tasks are learned before hard tasks. The key experiments primarily show that more general versions of a formatting constraint are learned before more specific versions of the same constraint. Such ordering is expected under incremental learning, independent of the particular optimization algorithm (for example, evolutionary methods will show the same effect).

**Missing definitions of difficulty and disjointness.**
For the findings to bear on "easy vs. hard," difficulty and disjointness must be formalized. The set $(r_1, r_2, r_3)$ is maximally non-disjoint (each stricter format is a subset of the looser one), so learning $r1 \rightarrow r2 \rightarrow r3$ may simply reflect learning shared substructure first. It might also reflect greedy optimization ("path of least resistance"), but the paper does not provide conceptual tools to distinguish these explanations.

A concrete proposal is to define difficulty $K(T)$ of a task T as the number of RL steps to reach a fixed success threshold. For two tasks with shared structure, it may hold that training on their sum takes roughly as long as training either alone. This motivates defining shared substructure as:
$ I(A:B) = K(A) + K(B) - K(A,B) $
This is an operational analogue of algorithmic mutual information.

**Ambiguity in the "disjoint" claim.**
The paper states that disjoint objectives are optimized, but does not specify which objectives. In the absence of a quantitative assessment of shared structure, it is unclear whether truly disjoint (structure-orthogonal) behaviors were tested or whether superficial variants (for example, minor stylistic changes) confounded the result.

**KL analysis adds little to the central claim.**
The KL plot (Figure 5, mean rewards) primarily shows that a high KL penalty inhibits learning, which is expected and tangential to the paper’s central question.

# Recommended reframing of the hypothesis and test

A cleaner statement would be: given tasks $A$ and $B$ with $K(A) < K(B)$ and $I(A:B) \approx 0$, joint training on $A$ + $B$ will solve $A$ before $B$. Let $K(B|A)$ denote the additional steps between solving $A$ and solving $B$. The key experiment is to vary $K(B) - K(A)$ (or the KL coefficient) while holding $I(A:B) \approx 0$, and measure the effect on $K(B|A)$. This would more fairly quantify the extent to which PO "prefers the path of least resistance."
**If you ran this experiment I would consider greatly increasing my score.**

**Questions:**

- What are the exact definitions/specifications of the three "disjoint" rewards used in Figure 4?

---

### Note · Authors · 2026-01-02

**Comment:**

We thank the reviewers for their time and the constructive reviews, which have vastly help us improve our work. We are withdrawing our paper from this cycle.

**Withdrawal Confirmation:**

I have read and agree with the venue's withdrawal policy on behalf of myself and my co-authors.